# Economic evaluation of the OSAC randomised controlled trial: oral corticosteroids for non-asthmatic adults with acute lower respiratory tract infection in primary care

Aida Moure-Fernandez,[1,2] Sandra Hollinghurst,[3] Fran E Carroll,[2] Harriet Downing,[3] Grace Young,[2] Sara Brookes,[4] Margaret May ![ORCID] ,[2,4] Magdy El-Gohary,[5] Anthony Harnden,[6] Denise Kendrick,[7] Natasher Lafond,[7] Paul Little,[5] Michael Moore,[5] Elizabeth Orton,[7] Matthew Thompson,[8] David Timmins,[6] Kay Wang,[6] Alastair D Hay[3]

For numbered affiliations see end of article.

**Correspondence to**
Aida Moure-Fernandez;
aidamoure@gmail.com

## ABSTRACT

**Objective** To estimate the costs and outcomes associated with treating non-asthmatic adults (nor suffering from other lung-disease) presenting to primary care with acute lower respiratory tract infection (ALRTI) with oral corticosteroids compared with placebo.

**Design** Cost-consequence analysis alongside a randomised controlled trial. Perspectives included the healthcare provider, patients and productivity losses associated with time off work.

**Setting** Fifty-four National Health Service (NHS) general practices in England.

**Participants** 398 adults attending NHS primary practices with ALRTI but no asthma or other chronic lung disease, followed up for 28 days.

**Interventions** 2× 20 mg oral prednisolone per day for 5 days versus matching placebo tablets.

**Outcome measures** Quality-adjusted life years using the 5-level EuroQol-5D version measured weekly; duration and severity of symptom. Direct and indirect resources related to the disease and its treatment were also collected. Outcomes were measured for the 28-day follow-up.

**Results** 198 (50%) patients received the intervention (prednisolone) and 200 (50%) received placebo. NHS costs were dominated by primary care contacts, higher with placebo than with prednisolone (£13.11 vs £10.38) but without evidence of a difference (95% CI £3.05 to £8.52). The trial medication cost of £1.96 per patient would have been recouped in prescription charges of £4.30 per patient overall (55% participants would have paid £7.85), giving an overall mean 'profit' to the NHS of £7.00 (95% CI £0.50 to £17.08) per patient. There was a quality adjusted life years gain of 0.03 (95% CI 0.01 to 0.05) equating to half a day of perfect health favouring the prednisolone patients; there was no difference in duration of cough or severity of symptoms.

**Conclusions** The use of prednisolone for non-asthmatic adults with ALRTI, provided small gains in quality of life and cost savings driven by prescription charges. Considering the results of the economic evaluation and

## Strengths and limitations of this study

► The economic evaluation was part of a rigorously conducted multicentre randomised controlled trial, involving a representative population of patients not thought to need immediate antibiotic treatment.
► The economic evaluation included the perspectives of patients and time off work as well as the National Health Service.
► Low levels of missing cost and outcome data, with EuroQol 5D observations from multiple time points, achieving an accurate profile of patient health-related quality of life over the period of the illness.
► The analysis was thorough and included multiple imputation of missing data and extensive sensitivity analysis.

possible side effects of corticosteroids, the short-term benefits may not outweigh the long-term harms.

**Trial registration numbers** EudraCT 2012-000851-15 and ISRCTN57309858; Pre-results.

## INTRODUCTION

Acute lower respiratory tract infection (ALRTI), with symptoms such as wheeze, phlegm and chest pain, is one of the most common reasons for patients to consult in primary care.[1] In the UK, ALRTI costs the National Health Service (NHS) at least £190 million annually,[2] and further costs are borne by patients in self-managing their condition[3 4] and by society in general because of work absenteeism.[5]

Despite lack of evidence of efficacy and the recommendation from the National Institute for Health and Care Excellence of a non-antibiotic pathway,[6] antibiotics are still

frequently prescribed to treat ALRTI.[7 8] This unnecessary prescribing fuels antimicrobial resistance, raises patient expectations for similar treatment in the future (so-called 'illness medicalisation') and is a wasteful use of healthcare resources.

Oral and inhaled corticosteroids are widely used to treat symptoms of asthma[9] and there is some evidence of the effectiveness of high doses of inhaled corticosteroids in reducing cough frequency among non-smokers with ALRTI who do not have asthma or other chronic lung disease.[10] Given the similarity in symptoms of ALRTI and those of asthma, we tested the hypothesis that corticosteroids might be an effective treatment for ALRTI in adults without asthma or other chronic lung disease. Indeed, there is increasing evidence from Europe[11] and the USA[12] of steroid prescribing for patients with ALRTI, with a secondary analysis of one US study[12] showing 15% of adults without asthma but with ALRTI being prescribed oral steroids.

The Oral Steroids for Acute Cough (OSAC) study[13 14] examined the effectiveness and costs of oral corticosteroids (more specifically oral prednisolone) in treating ALRTI in a non-asthmatic adult population compared with placebo. Here, we report on the results of the economic evaluation.

## METHODS

The OSAC study was a two-arm randomised controlled trial that aimed to test the provision of oral corticosteroids to adults with ALRTI but no asthma or other chronic lung disease such as chronic obstructive pulmonary disease. Details of the trial procedures and clinical results are reported elsewhere.[13 14] Briefly, patients with an ALRTI-related cough were recruited from 54 primary care practices across four areas of England and randomly allocated to receive either oral prednisolone (intervention) or placebo (control) for their acute cough symptoms. The intervention consisted of 40 mg (two 20 mg tablets) of prednisolone daily for 5 days. The control group received matching placebo tablets. The primary clinical outcomes of the study were duration of 'moderately bad or worse' cough and symptom severity for days 2 to 4 postrandomisation. The primary outcome for the economic evaluation was quality adjusted life years (QALYs) which were estimated from responses to the 5-level EuroQol-5D version (EQ-5D-5L).[15]

### Design

The economic evaluation was conducted from the perspectives of the NHS, patients and a broader perspective taking into account differences in work-related productivity costs; we included all resource use during the 28-day follow-up period. We used a cost-consequences design,[16] comparing cost from the three perspectives with the clinical outcomes and QALYs to see all possible costs and health consequences.

### Identification of resources

NHS resources were identified as being: the trial medication, primary care consultations, other relevant prescribed medication, hospital care, use of NHS 111 and ALRTI-related investigations such as chest X-ray. Patient resource use was identified as being: travel, prescription costs and over-the-counter medication and remedies. Productivity losses were identified as being days off work due to the cough.

### Measurement of resources: data collection

A primary care notes review took place at the end of the study period, and data were extracted on: all primary care consultations, categorised as doctor or nurse, face to face, telephone or home visit and in-hours or out-of-hours; prescribed medication and investigations. Participants in the study were issued with a daily diary to complete, which was used to collect data on health service use not always reliably available from primary care notes such as use of NHS 111 and hospital services. Participant out-of-pocket expenses on travel and over-the-counter medicines were also recorded here, along with time off work. Participants were telephoned weekly to reinforce their record keeping.

### Valuation of resources

Resources were valued as shown in table 1. All resources were costed in pounds sterling at 2013–2014 prices, using an appropriate inflation index[17] when necessary. Costs for primary care services were obtained from Curtis[17]; calls to the NHS 111 service were costed using a published national evaluation,[18] and NHS reference costs[19] were used to value secondary care services. Travelling by car was valued using the Automobile Association (AA) running schedule to cost the mileage,[20] and standard ticket prices were employed to cost the use of public transport. Productivity costs were derived using the age/sex average rate from the Annual Survey of Hours and Earnings.[21]

For prescribed medication, costs were extracted mainly from the British National Formulary (BNF).[22] A cost was obtained for each prescription based on the name of the drug, the method of administration (for example, tablet, capsule, liquid) and the number of units included in the prescription. When the cost of a drug was not available in the BNF, the Prescription Costs Analysis of England database[23] was used to cost that medication. Uncertainties around relevance (to ALRTI) and missing information were resolved by seeking clinical advice.

Careful consideration was given to the most accurate way of costing the intervention medication (prednisolone) with the aim of reflecting the true cost to the NHS if it were to be adopted as a strategy for treating ALRTI. First, thought was given to the number of pills that would be needed to cover the intervention total dose as the 20 mg tablets provided in the study are not routinely available. We assumed that 5 mg tablets would be a reasonable substitute, resulting in 8 tablets required per day, and 40 tablets for the duration of the intervention

**Table 1** Valuation of resource use

| Category of resource | Unit cost 2013/2014 |
|---|---|
| Primary care services | |
| GP (face to face in practice) | 46.00* |
| GP (telephone call) | 28.00* |
| GP (out of hours) | 68.30* |
| Practice nurse (face to face in practice) | 13.69* |
| Practice nurse (telephone call) | 9.00* |
| Nurse practitioner(face to face in practice) | 25.00* |
| Nurse practitioner (out of hours) | 33.41* |
| NHS 111 calls | 8.29† |
| Hospital and walk-in services | |
| Walk-in centre | 40.50‡ |
| Outpatient | 150.00/214.00§¶ |
| A&E visits | 111.20‡ |
| Investigations (X-ray) | 28.01‡ |
| Medication | |
| Prescribed medication | By item§** |
| Prednisolone (intervention) | 1.96§** |
| Over-the-counter medication | By item |
| Prescription charge | 7.85 |
| Other | |
| Mileage | 0.64†† |
| Time off work | 118.24‡‡ |

*Curtis,[17]
†Evaluation report.[18]
‡NHS reference costs.[19]
§BNF.[22]
¶£214.00 was used for a patient who had a very resource-intensive outpatient stay (upper bound outpatient cost).
**Prescription Pricing Authority.[23]
††AA schedule.[20]
‡‡Office of National Statistics.[21]
A&E, accident and emergency; BNF, British National Formulary; GP, general practitioner; NHS, National Health Service.

(5 days). From the basic price of £1.03 given in the BNF, adjustments were made as follows. We subtracted the usual discount (7.61% of the basic price) eligible to the dispensing pharmacy from the manufacturer; we added a standard 90p dispensing fee and a payment of 1.24p for consumables; and as the amount required was 40 and packs contain 28 tablets, an extra 10p for splitting packs was added. This resulted in a cost for the intervention prednisolone of £1.96. To reflect a 'roll-out' situation, we allowed for potential receipts from prescription payments made by patients by including an amount for those participants who reported that they usually pay a prescription charge.

No cost was included for the placebo medication (ie, tablets) as this was a research cost.

## Measurement and valuation of outcomes

The EQ-5D-5L was completed weekly by participants from baseline during the 28-day follow-up, giving us five observations to use to form QALYs. Utilities were obtained from existing preferences elicited from the general public, using the cross-walk algorithm.[15] QALYs were calculated from these utilities using the area under the curve and adjusting for baseline differences.[24]

Results from the clinical trial[14] show that the median duration of moderately bad or worse cough was 5 days in both groups and the HR of 1.11 (95% CI 0.89 to 1.39) also indicates no difference. There was a reduction of 0.02 in mean symptom score for days 2 to 4 in prednisolone patients though this was less than the predetermined clinically meaningful difference.

## Data analysis

Frequencies of resource use were calculated to provide a descriptive analysis of the types of resources used by primary care patients with a cough. We estimated mean resource use and cost per patient for all categories, by trial arm, to compare the prednisolone group with those using the placebo. Ordinary least squares regressions were employed to calculate the differences in costs adjusted for centre, age, gender and outcome-related baseline variables to account for potential imbalances between the groups. Standard deviations (for means) and bias-corrected bootstrapped CIs[25] (2000 replicates) were constructed to account for the uncertainty in the point estimates.

No discount was applied to the data as the time horizon of the study was 28 days. All analyses were carried out using Stata V.13 and above.[26]

## Sensitivity analyses

Five different sensitivity analyses were conducted to test the robustness of assumptions made in the base case analysis. The effect of missing data was appraised in two scenarios using multiple imputation techniques: in addition to the imputation with chained equations (ICE) (scenario 1), which is usually employed in economic evaluations, an analysis was also carried out using the 'twofold' command in Stata; this command imputes missing values at a given time point, conditional on information at the same time point and immediately adjacent time points[27] (scenario 2), to be consistent with the methodology used in the clinical effectiveness analysis.[14] Scenario 3 excluded outpatient and accident and emergency attendances where there was any ambiguity about their relevance to LRTI; scenario 4 used the visual analogue scale to calculate QALYs; and scenario 5 removed the prescription payments from the analysis on the basis that these are to some extent artefactual and relate specifically to England at the time of the study.

## Patient and public involvement

Patient participation and involvement (PPI) input was important in the decision to prioritise the research

question for funding, and their views informed discussions around the design of the study, the sample size calculation and selection of primary and secondary outcomes. PPI views were sought on recruitment methods, and all patient facing trial materials. The burden of the intervention was also assessed by patients themselves, and results were disseminated to both practices and the patients. We wish to thank our PPI advisors for their input into trial design and management.

## RESULTS

In total, 398 patients were included in the intention-to-treat analysis. One hundred and ninety-eight (49.7%) were in the trial medication (intervention) group and 200 (50.3%) received the matched placebo (control). Baseline characteristics of the participants are shown in online supplementary appendix table A. Participants were predominantly white, employed and middle aged and similar rates received asthma medication in both groups (5% vs 4%) during the previous years. Approximately 50% in both arms stated they had never smoked. Data on NHS costs were complete for 332 (83%) participants and out-of-pocket costs for patients were reported by 329 (83%). Time off work was recorded by 321 (81%) of the participants filling the questionnaire and 346 (87%) completed the EQ-5D-5L at all five time points.

### Resource use

Use of all health services was relatively modest during the 28-day follow-up period. Overall just 20% participants accessed primary care, a proportion that was similar for both groups though patients in the placebo group had slightly more consultations per patient than those in the prednisolone group, which is reflected in the mean number of encounters (0.30 vs 0.27) as shown in table 2. Prescribed medication was also slightly higher in the placebo group in terms of number of patients (15% vs 12%) being prescribed anything and the mean number of prescriptions per participant (0.22 vs 0.15). Nine participants (2%) reported using

hospital services: three (1.5%) in the prednisolone group and six (3%) in the placebo group.

More placebo group participants reported buying over-the-counter medications than those in the prednisolone group (45% vs 39%), and the mean number of items per participant was higher (0.59 vs 0.48). Popular items included cough linctus and cold and influenza remedies. Similarly, more placebo group participants reported time off work than prednisolone participants (32% vs 28%), and the mean length of time off was longer (1.38 days vs 0.95 days).

Although there was a consistently higher use of resources in the placebo group compared with the prednisolone group, there is imprecision around the point estimates and, with the exception of the trial medication and associated prescription payments, there is no evidence of a difference in use between the groups as indicated by the CIs reported in table 2.

### Cost analysis

Table 3 summarises the cost comparison between the two groups. The majority of NHS costs were attributable to primary care consultations with hospital visits, including for X-rays, contributing a modest amount. Over half (55%) of prednisolone participants reported that they normally pay a prescription charge; the value of prescription payments made by these patients more than covered the cost of the trial medication meaning that the NHS on balance would have made a profit. Out-of-pocket patient costs were dominated by prescription payments and the value of time off work was higher than any other category of cost. Comparing the prednisolone and placebo groups, costs were higher in the placebo group for all categories (except those related to the trial medication) though again there was no evidence of a difference between the groups as indicated by the CIs.

### Cost-consequence analysis

Table 4 shows the comparison of incremental costs and outcomes, including the results from the clinical trial regarding duration of cough and symptom severity score.

| Table 2 | Mean (SD) resource use, per patient, by category and group (all available data) | | | | |
|---|---|---|---|---|---|
| | **Prednisolone** | | **Placebo** | | **Unadjusted difference** |
| **Resource use category** | **n** | **Mean (SD)** | **n** | **Mean (SD)** | **Mean (95% CI)** |
| Primary care consultations | 195 | 0.27 (0.58) | 198 | 0.30 (0.73) | −0.03 (−0.16 to 0.10) |
| Prescribed medication | 190 | 0.15 (0.51) | 200 | 0.22 (0.64) | −0.06 (−0.18 to 0.05) |
| NHS 111 calls | 177 | 0.04 (0.22) | 168 | 0.02 (0.13) | 0.02 (−0.02 to 0.06) |
| X-ray procedures | 197 | 0.04 (0.19) | 198 | 0.04 (0.20) | 0.00 (−0.04 to 0.03) |
| All hospital visits | 176 | 0.02 (0.13) | 166 | 0.04 (0.23) | −0.03 (−0.06 to 0.01) |
| Trial medication | 198 | 1 (0) | 200 | 0 (0) | 1 (1 to 1) |
| Prescription payments | 191 | 0.62 (0.66) | 181 | 0.09 (0.39) | 0.52 (0.41 to 0.64) |
| Over the counter medication | 180 | 0.48 (0.67) | 168 | 0.59 (0.83) | −0.11 (−0.26 to 0.05) |
| Time off work (days) | 171 | 0.95 (2.61) | 165 | 1.38 (3.05) | −0.43 (−1.04 to 0.18) |

NHS, National Health Service.

**Table 3** Mean (SD) cost (£) by group, per patient, by category and group (all available data)

| Resource use category | Prednisolone | | Placebo | | Unadjusted difference |
| --- | --- | --- | --- | --- | --- |
| | n | Mean (SD) | n | Mean (SD) | Mean (95% CI) |
| Primary care consultations | 195 | 10.38 (23.97) | 198 | 13.11 (33.52) | –2.73 (–8.52 to 3.05) |
| Prescribed medication | 195 | 0.36 (1.27) | 194 | 0.44 (1.33) | –0.08 (–0.34 to 0.17) |
| NHS 111 calls | 168 | 0.33 (1.85) | 177 | 0.33 (1.85) | 0.18 (–0.14 to 0.50) |
| X-ray procedures | 198 | 1.00 (5.20) | 197 | 1.00 (5.20) | –0.14 (–1.20 to 0.93) |
| All hospital visits | 176 | 1.09 (9.39) | 166 | 3.84 (23.10) | –2.75 (–6.46 to 0.97) |
| Trial medication | 198 | 1.96 (0) | 200 | 0 (0) | 1.96 (1.96 to 1.96) |
| Prescription payments* | 191 | 4.85 (5.20) | 181 | 0.74 (3.06) | 4.11 (3.24 to 4.99) |
| Over the counter medication | 175 | 2.23 (3.93) | 165 | 3.08 (5.07) | –0.85 (–1.82 to 0.11) |
| Travel costs | 171 | 1.76 (2.63) | 166 | 2.45 (4.39) | –0.69 (–1.47 to 0.08) |
| Time off work (days) | 161 | 58.02 (161.16) | 160 | 83.88 (183.93) | –25.86 (–63.83 to 12.11) |

*Prescription payments are a transfer cost between the NHS and patients. They are a negative cost (receipt) to the NHS and a positive cost to patients.
NHS, National Health Service.

Here, we present the difference in cost by perspective, adjusted by centre, age, gender and baseline covariates. The 95% CIs are bootstrapped and bias corrected. The negative incremental cost of –£7.00 (95% CI –£17.08 to –£0.50) per patient to the NHS indicates the intervention is cheaper than placebo; it reflects the lower use of primary care by patients in the prednisolone group and the offset of intervention costs by prescription payments. From the patient perspective, despite higher travel and over-the-counter costs in the placebo group, total costs were approximately £3 more in the prednisolone group due to the cost of prescriptions. The value of time off work was approximately £30 higher per patient for the placebo group.

The gain in quality of life provided by the prednisolone was 0.03 QALYs (95% CI 0.01 to 0.05) per patient, which translates into slightly more than half a day of extra 'best imaginable' health during the 28 days. The percentage of patients reporting no problems for each domain of the EQ-5D are shown in online supplementary appendix table B. On average, the prednisolone patients improved by more than the placebo patients in all domains, but the 'pain and discomfort' and 'usual activities' domains were where the greatest difference was seen.

### Sensitivity analysis
Results of the sensitivity analyses are presented in table 5. Imputing missing data and excluding unrelated costs make no difference to the conclusions of the base case analysis. Using the visual analogue scale to value the QALYs reduced the QALY gain of prednisolone patients over placebo to a minimal amount (0.01: 95% CI –0.01 to 0.03). When prescription payments are removed, the incremental cost to the NHS is still negative (–£3.04; 95% CI –£11.31 to £5.24) though the CI suggests no evidence of a true difference. The reverse is true for patient costs as placebo patients spent considerable more on travel and over-the-counter medications and remedies than those in the prednisolone group.

### DISCUSSION
#### Summary of main findings
Prednisolone was found to be clinically ineffective for treating ALRTI in non-asthmatic adults in primary care in terms of duration of cough and symptom severity[14]; however there was evidence that patients using prednisolone experienced a greater improvement in health-related quality of life than those using the placebo. This was largely due to a greater improvement in pain/discomfort and a speedier return to carrying out usual activities. Prednisolone is a relatively inexpensive medication and, in this population of generally healthy adults ineligible for prescription charge exemption, prescription payments more than offset the cost to the NHS. Other NHS resource use was consistently higher in the placebo group across all categories though the differences were small. The value of time off work was considerable, and this was higher in the placebo patients,

**Table 4** Cost-consequence analysis. Differences in mean (95% CI) cost and QALY (complete cases, by perspective)

| | Mean (95% CI)* difference |
| --- | --- |
| All NHS services (n=332) | –£7.00 (–£17.08 to –£0.50) |
| All patient out-of-pocket expenditure (n=329) | £2.90 (£1.14 to £4.48) |
| Value of time off work (n=321) | –£30.45 (–£ 67.15 to £9.79) |
| QALYs (n=346) | 0.03 (0.01 to 0.05) |
| Duration of cough (n=334) | HR: 1.11 (0.89 to 1.39) |
| Symptom severity score (n=368) | –0.20 (–0.40 to 0.00) |

* Biased corrected and adjusted by centre and baseline covariates.
NHS, National Health Service; QALYs, quality adjusted life years.

**Table 5** Sensitivity analyses: incremental costs and QALYs under different scenarios

| Scenario | Differences in means* (95% CI) |
|---|---|
| **1. Imputed data (ICE)** | |
| NHS services | −£9.78 (−£19.05 to −£3.87) |
| Patient out-of-pocket expenditure | £3.01 (£1.06 to £4.28) |
| Value of time off work | −£22.99 (−£55.81 to £12.97) |
| QALYs | 0.03 (0.01 to 0.04) |
| **2. Imputed data (twofold)** | |
| NHS services | −£7.27 (−£17.12 to −£0.91) |
| Patient out-of-pocket expenditure | £2.97 (£1.16 to £4.54) |
| Value of time off work | −£28.36 (−£67.75 to £9.38) |
| QALYs | 0.03 (0.01 to 0.04) |
| **3. Exclusion of potentially unrelated cost categories (n=332)** | |
| NHS services | −£6.20 (−£13.77 to £0.14) |
| **4. QALYs using VAS (n=326)** | |
| QALYs | 0.01 (−0.01 to 0.03) |
| **5. Removal of prescription payments (n=334)** | |
| NHS services | −£3.04 (−£11.31 to £5.24) |
| Patient out-of-pocket expenditure | −£1.49 (−£2.83 to −£0.16) |

*Adjusted by centre and baseline covariates.
NHS, National Health Service; QALYs, quality adjusted life years; VAS, visual analogue scale.

though the limited sample size combined with patient-level variation prevented a robust conclusion.

### Strengths and weaknesses

The sample included in OSAC was typical of patients consulting in primary care with ALRTI, not thought to require immediate antibiotic treatment. The two groups were well matched at baseline and there was minimal drop-out. The economic evaluation was carried out at individual patient level and covered the whole 28-day period. Data on actual utility values for each group was not reported, which would have helped to understand the differences regarding quality of life data between each group; however, the weekly collection of EQ-5D-5L data allowed for an accurate estimate of the recovery profile of patients, capturing the speedier improvement of those in the prednisolone group, suggesting that they 'felt better' more quickly than those taking the placebo. In this trial, there were low levels of missing data; however, when estimating cost at a number of time points over the whole trial period, any missing resource use data pose a threat. From the NHS perspective, there were 332 complete cases representing 83% of the total sample who provided data at all time points. The total amount of missing data points was less than this which is borne out by the results of the multiple imputation, confirming that the complete cases estimates are robust.

The size of the sample for the trial was based on the primary clinical outcomes, as is common practice. Patient variability affected the uncertainty around the estimates of mean cost, so although there were suggestions of differences between the groups (especially for time off work), the sample was not large enough for this to be confirmed beyond chance. The economic evaluation was restricted by the procedures and time frame of the trial. We did not include a cost for the placebo medication, but it is possible that in a 'real-life' situation, an alternative treatment such as a codeine linctus, an inhaler or an oral antibiotic might be prescribed. Although it was appropriate to limit the follow-up period to 28 days to answer the clinical question, we were unable to capture any long-term effects of either repeated use of corticosteroids, such as illness medicalisation or osteoporosis,[28] or a reduction in repeated days off work.

### Comparison with other literature (strengths and weaknesses)

We are aware of no other studies investigating the clinical and economic implications of corticosteroids for ALRTI. However, our estimates of cost and quality of life can be compared separately with other literature. Oppong *et al*[29] conducted a thorough analysis of resource use and cost for acute cough/LRTI in 13 European countries. The use of primary healthcare services in that study was slightly higher than our figure of 0.28 per participant: they found a mean of 0.34 visits for the two UK centres included in the study and an overall range of 0.30 to 1.89. However, this difference is small and may be accounted for by different inclusion criteria. On the other hand, the amount of time off work was considerably different. Our participants reported a mean length of 1.16 days off work compared with 3.08 in the European study, though it is difficult to tell whether those authors included a value (representing 'potential' time off) for participants not in paid work. Only one-third of our participants reported any time off work, but for those who did, the mean length was 3.9 days. In this study, we found a small but significant improvement in quality of life for patients taking prednisolone compared with those taking the placebo. In a recent review of short course oral steroids for chronic rhinosinusitis,[30] an improvement in quality of life was observed at the end of a 2–3-week course though the quality of this evidence was judged to be low. There is also some evidence of patients reporting positive mood change when taking steroids.[31]

### Meaning of the study/policy implications

The results of this study pose an interesting challenge in terms of interpretation. Despite the negative clinical (cough duration and symptom severity) endpoint results, we found prednisolone was cheaper (in terms of NHS costs) than placebo and better (in terms of QALY gain), but each of these findings comes with a caveat. Once the benefit of prescription payments was removed from the analysis the cost gain was much reduced, and could have been due to chance. Placebo patients consistently used more healthcare services than those on prednisolone, but as the trial was not powered to detect a difference in cost, we cannot draw any

firm conclusions. Although the short-term use of prednisolone may make patients feel a little better, the negative effects of longer term or repeated use, such as illness medicalisation[32] and osteoporosis[28] cannot be accounted for in this analysis and are potentially powerful arguments against more widespread use.

The results of this study evidence a small increment of the quality of life—particularly in pain/discomfort and resumption of usual activities—in patients who take prednisolone to treat their ALRTI symptoms. However, taking the clinical and economic implications derived from this study, and considering the possibility of side-effects from repeated short-term use, the benefits may not outweigh the unknown long-term risk.

### Unanswered questions and future research

This study has addressed the question of the short-term use of corticosteroids for ALRTI, but more work is needed to understand the longer term influences. One motivation for carrying out the research was concern about inappropriate prescribing of antibiotics for conditions such as ALRTI, particularly with respect to the effect on antimicrobial resistance. Corticosteroids are successfully used in the short term for many conditions and the long-term detrimental effects, such as osteoporosis, high blood pressure and suppression of the immune system are recognised; a recent study[33] showed that adverse events can develop within 1 month of short-term steroid use in patients with acute respiratory tract infections; however, the specific long-term effect of repeated short-term use of steroids by ALRTI patients is uncertain. Future work could include long-term modelling of the costs and effects of alternative treatments for ALRTI, including corticosteroids, antibiotics and other medications with little or marginal effect such as linctus. The modelling could take a broad perspective, including the impact of lost productivity, and address the challenge of weighing up the societal cost of antimicrobial resistance against patient level side-effects of medication and the potential for overmedicalisation of ALRTI. In some studies,[2 34] the small gains in time to recovery provided by antibiotics (12–24 hours) was not sufficient to outweigh their risks. Our study did not measure this variable; however, future research would be necessary to ascertain whether a similar conclusion could be obtained for prednisolone.

### CONCLUSION

The economic evaluation evidences gains in quality of life provided by the use of prednisolone for non-asthmatic adults with ALRTI from the perspective of the NHS. However, the benefit is small and taking the clinical and economic implications derived from this study and considering the potential side effects from repeated use of corticosteroids, the short-term benefits may not outweigh the long-term harms.

**Author affiliations**
[1]Bristol Randomised Trials Collaboration, University of Bristol, Bristol, UK
[2]School of Social and Community Medicine, University of Bristol, Bristol, UK
[3]Centre for Academic Primary Care, University of Bristol, Bristol, UK
[4]Population Health Sciences, University of Bristol, Bristol, UK
[5]Department of Primary Care and Population Sciences, University of Southampton, Southampton, UK
[6]Nuffield Department of Primay Care Health Sciences, University of Oxford, Oxford, UK
[7]Division of Primary Care, University of Nottingham, Nottingham, UK
[8]Department of Family Medicine, University of Washington, Seattle, Washington, USA

**Acknowledgements** We wish to thank participants, the recruiting primary care sites, the Clinical Research Networks and all members of the OSAC team. We would also like to thank the Trial Steering and Data Monitoring Committee members, Nottingham University Hospitals NHS Trust pharmacy and the University Hospitals Bristol NHS Foundation Trust.

**Contributors** AH was the Chief Investigator of the OSAC trial; SH and FEC designed the economic evaluation; AMF carried out the economic analysis under the supervision of SH. HD was trial manager; MaM designed the statistical analysis which was carried out by GY and SB. AM-F, SH, FEC HD, GY, SB, MaM, ME-G, AH, DK, NL, PL, MiM, EO, MT, DT, KW and ADH contributed to the interpretation of the results. AMF wrote the first draft of the paper. SH, FEC, HD, GY, SB, MaM, ME-G, AH, DK, NL, PL, MiM, EO, MT, DT, KW and ADH reviewed the manuscript and approved the final version.

**Funding** This work was supported by the NIHR School for Primary Care Research (grant reference 117a). ADH was funded by NIHR Research Professorship (NIHR-RP-02-12-012). A trial steering committee provided independent supervision on behalf of the funder and sponsor (University of Bristol) and an independent data monitoring committee oversaw safety.

**Disclaimer** The views expressed are those of the authors and not necessarily those of the NHS, the NIHR or UK Department of Health.

**Competing interests** All authors have completed and submitted the ICMJE Form for Disclosure of Potential Conflicts of Interest. Dr Thompson reports that he has received funding from Alere Inc to conduct research on C-reactive protein point-of-care tests, has received funding from Roche Molecular Diagnostics for consultancy work.

**Patient consent for publication** Not required.

**Ethics approval** Ethical approval was granted by the Central Bristol Research Ethics Committee (12/SW/0180).

**Provenance and peer review** Not commissioned; externally peer reviewed.

**Data availability statement** Data are available upon reasonable request. All the OSAC trial data are available on request and only for academic purpose. Please contact Mrs Grace Young (grace.young@bristol.ac.uk) for data requests.

**ORCID iD**
Margaret May http://orcid.org/0000-0002-9733-1003

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
