## [Reviewer comments · BMJ Open]

ARTICLE DETAILS

TITLE (PROVISIONAL)	Economic evaluation of the OSAC randomised controlled trial: oral corticosteroids for non-asthmatic adults with acute lower respiratory tract infection in primary care
AUTHORS	Moure-Fernandez, Aide; Hollinghurst, Sandra; Carroll, Fran; Downing, Harriet; Young, Grace; Brookes, Sara; May, Margaret; El-Gohary, Magdy; Harnden, Anthony; Kendrick, Denise; Lafond, Natasher; Little, Paul; Moore, Michael; Orton, Elizabeth; Thompson, Matthew; Timmins, David; Wang, Kay; Hay, Alastair

VERSION 1 – REVIEW

REVIEWER	C.H. van Werkhoven University Medical Center Utrecht, the Netherlands
REVIEW RETURNED	25-Sep-2018

GENERAL COMMENTS	The authors performed a cost-consequence analysis using data from the OSAC trial: a randomised comparison of a 5 day course of predinolone to placebo in adults with acute lower respiratory tract infection in primary care. The manuscript is well written and informative, and the study is well conducted both in terms of patient follow-up, data collection, and data analysis. The discussion is balanced in not overemphasizing statistically significant results but really looking at the relevance of the observed effects, which I appreciate. I do have some (mostly small) questions and suggestions for improvement which are listed below. Main comments/questions:  - I do not understand the QALYs value and the translation to a difference of "half a day in best imaginable health". A period of 28 days in perfect health makes a QALY of 0.077 (1/365*28) so this is the maximum value of each participant. 0.03 QALY would translate to 11 days in perfect health (0.03*365). Otherwise stated, over the 28 day period, their quality of life is, on average, almost 40% better (absolute difference). This would be a major health improvement but I cannot imagine it is true! So please check the QALY calculation or enlighten this reviewer, maybe my own calculations are wrong. - For scenario 5 (NHS perspective) and for the basecase scenario (patient perspective) it would add value if a willingness to pay curve was added to reveal what is a likely value of the cost per QALY gained. Many patients are able and willing to pay some money to recover faster, and in general society is willing to invest in QALY gain. - The authors performed a cost-consequence analysis. The difference with a conventional cost-effectiveness analysis is - as I understand it - that not only (quality of) life years gained but also
---

other (clinical) consequences are reported. In this case, the duration of cough and the symptom severity score at day 2-4 post randomisation are reported. However, these outcomes were already reported in Hay et al. JAMA 2017, so although it might add value to report them again, it should be clearly stated in the abstract, methods and results section (including the tables) that these have been reported previously. If any difference, it should be clarified where they arise from.

abstract:

- In objective 'non-asthmatic adults' but in participants 'no asthma or other chronic lung disease'. Please use 'no asthma or other chronic lung disease' consistently.
- outcome measure: please include the time window of 28 days for the outcomes.
- "the benefit in terms of quality of life is small" -> did the authors have higher expectations in this population? Even if one would feel better two days earlier, that would probably still translate to "half a day in perfect health" (assuming severe cough would roughly reduce ones quality of life by 25%). What benefit would be large enough to justify one course of corticosteroids?
- In outcome measures: costs should also be included.
- In strengths and limitations: 'The economic evaluation included the perspectives of patients and time off work as well as the NHS.' But from the abstract it seems the cost evaluation is only from a healthcare perspective.

Methods:

- Page 8 line 28: please make clear that the EQ-5D-5L was also measured at baseline, maybe repeat the 28 day follow-up period.
- "Stata 13 and above" do you mean that you used higher versions as well? Please specify which versions.
- Please clarify whether "Prescription charge" is also included in the costs of non-study medication. (I think it should be.)

Results:

- Page 10 line 10: Time off work was recorded by 321 (81%) participants. The wording might be clarified to avoid readers thinking the other participants were unemployed, or had zero work absenteeism, or did not complete this question. I assume the latter is meant, if so it might clarify if it is explicitly stated that 19% did not complete the question on work absenteeism.
- Page 10 line 11: Maybe to add what proportion had zero follow-up EQ-5D-5L questionnaires completed as this would severely complicate the area under the curve estimation.
- page 12 line 22: "heath" should be "health".
- page 12 line 35-42: similar to previous comment, these data have already been reported. As they are not an integral part of the cost-consequence analysis (or else I have missed it) they should not be repeated. Of course they can be discussed in the discussion.
- Table 4: Please also add summary statistics per intervention group. Please clarify that a HR > 1 means a shorter duration of cough. Please include the day of the symptom severity score.
- Table 5: Number of patients in scenario 5 has dropped off.
- It would be valuable to have a detailed description of the other prescriptions provided. E.g. if prescribing prednisolone would prevent prescription of antibiotics it might be more worth it, despite the small difference in clinical outcome.
- It would add insight if the QALY over the 28 day period would be plotted for the two groups.

	Discussion: - "Prednisolone was found to be clinically ineffective for treating ALRTI in non-asthmatic adults in primary care" Please make clear that this statement is about cough duration and symptom severity after 2-4 days (and maybe refer to the paper by Hay et al.).
--	--

REVIEWER	Cynthia Gong Children's Hospital Los Angeles University of Southern California United States
REVIEW RETURNED	29-Oct-2018

GENERAL COMMENTS	The authors have done a thorough job accounting for all possible costs associated with the patient perspective, including productivity and transport. These are typically lacking in most economic evaluations. The authors have also done a good job presenting different perspectives (i.e. NHS vs. patient). It is always helpful to see how a clinical trial translates into economic benefits. However, there are several issues that must be addressed for this analysis to be fit for publication. 1) This is not a cost-effectiveness evaluation; it is a cost-consequence evaluation. The paper should be retitled as such to avoid any confusion. Secondly, the authors do not provide any rationale as to why a cost-consequence evaluation was chosen over a cost-effectiveness evaluation. The paper by Coast was cited, but the latest panel of cost-effectiveness does not advocate the methodological solution described in the Coast paper, nor has her approach been vetted as a scientifically rigorous method for economic evaluation, so there needs to be better justification as to why the authors chose this approach. 2) It is unclear how 0.03 QALYs translates into half a day as there is no detailed information in the methods as to how the EQ5D assessments, done on a weekly basis, were then translated into QALYs. The authors should provide an explicit example in the methods of how QALYs were calculated using AUC, and specify how the time horizon of 28 days was taken into account, and how 0.03 QALYs then translates into half a day. If the measurement difference is indeed 0.03 QALYs, then $0.03 * 365 \text{ days} = 10.95 \text{ days}$, not half a day. If there are space limitations, perhaps this could be included in the supplemental section. 3) Is there a reason no probabilistic sensitivity analysis was included? While the scenario analyses address some certainty associated with the base model, there is no test of uncertainty within each parameter used to generate cost and QALY estimates. Although the data are from a clinical trial, there needs to be uncertainty assessment of each parameter so that decision-makers considering this information in clinical practice can understand how different rates of utilization may affect the overall costs and consequences. The authors themselves state in the Discussion that "...the limited sample size combined with patient-level variation prevented a robust conclusion"; this is where a probabilistic sensitivity analysis would be helpful to characterize the amount of patient variation. 4) It would be helpful to include a summary of the clinical trial results in the Methods section, first paragraph. Were statistical analyses done to control for smoking status and/or asthma / allergic rhinitis? 5) Were prescription costs for the placebo group included? Page 8, last sentence of top paragraph, says no cost was included
--

	for placebo medication. Are the authors referring to just the placebo tablets? If so, then please relabel placebo medication as placebo tablets, as “medication” implies therapeutic effect, which is not the case here. If placebo patients got prescription meds, those costs should be included, unless the number and type of prescription meds was identical between the treatment and control groups. 6) What “prescription medications” did patients get in addition to prednisolone / placebo? Did those in the placebo group get more rx medication because they went back for a repeat visit after their symptoms resolved? Nowhere do the authors discuss the other prescribed medications, or why only prednisolone was costed so precisely; was there no discount price and dispensing fee included in the cost of other rx medications? 7) Table 2: why is there only a mean of 0.27 primary care consultations in the treatment group? Isn’t a primary care consult required in order to get a prescription for prednisolone? Shouldn’t the mean be closer to 1.0? 8) The authors should include a table of the actual utility values, or at least a summary by week, comparing the two groups (a figure may be appropriate as well). This would help readers understand what (and when) is driving the differences in QALYs between each group. Table B provides no insight into the factors driving the differences between the groups. 9) It would be useful to include p-values in supplemental table B to compare whether there were significant differences between the two groups as far as % of patients reporting no problems. 10) Can the authors provide additional literature quantifying how the magnitude of their QALY estimates compare to other studies that look at similar populations? For example, although these patients were not necessarily on corticosteroids, the papers by Bergus et al, 2008, and Johnson FR et al, 2000, include QOL utility estimates that reflect respiratory illness and improvement in symptoms given treatment. Comparing the authors’ results to these papers would provide better context for whether the results of the current study make sense.
--	--

REVIEWER	Claudine A. Blum Medical University Clinic, Kantonsspital Aarau, Switzerland
REVIEW RETURNED	10-Dec-2018

GENERAL COMMENTS	This is a cost-consequence analysis of an RCT investigating 40 mg prednisolone vs. placebo in ALRTI. The manuscript is sound and well-balanced. Interestingly, this cost-consequence and patient-wellbeing analysis (also covers patients’ wellbeing through the EQ5D questionnaire), has “positive” results through half a day earlier reached wellbeing in the prednisolone group, whereas the endpoints of the main manuscript were not significantly different (duration to resolving symptoms such as cough). I had to go back to the main manuscript to find this out. I therefore encourage the authors to highlight the difference between the main results of the RCT and this “alongside” analysis, i.e. predefined secondary analysis, in the discussion. Furthermore, I invite the authors to be more specific at the beginning of the discussion (page 13 line 21) : they state that prednisolone has been found to be clinically ineffective in ALRTI. They should state that they refer to the main manuscript of the RCT (please also include the reference there).
---

	The authors choose to be cautious with the results and state in the conclusion that the harms of the trial may outweigh the shown benefits, even though they cannot pin down the extent of harms. I agree with the authors that the long-term risks of a “positive” glucocorticoid trial in a common disease should be highlighted, as it may result in a widespread frequent use of glucocorticoids. This main caveat has been well-covered by the authors. However, I encourage the authors to positively conclude the results of this analysis in the abstract, namely that the evaluation of economic aspects and patient-wellbeing resulted in a slight benefit “pro” glucocorticoids, before adding the caveat of potential harm. I would like to encourage the editors to have a statistician review the paper.
--	---

VERSION 1 – AUTHOR RESPONSE

Reviewer 1: Gong (California)

This is not a cost-effectiveness evaluation; it is a cost-consequence evaluation. The paper should be retitled as such to avoid any confusion. Secondly, the authors do not provide any rationale as to why a cost-consequence evaluation was chosen over a cost-effectiveness evaluation. The paper by Coast was cited, but the latest panel of cost-effectiveness does not advocate the methodological solution described in the Coast paper, nor has her approach been vetted as a scientifically rigorous method for economic evaluation, so there needs to be better justification as to why the authors chose this approach.

Despite this is a cost consequence analysis, this type of analyses are part of the cost-effectiveness analysis umbrella. We thought it would be better to specify CE in order to be more visible and then specify the cost-consequence definition in the abstract and throughout the text.

It is unclear how 0.03 QALYs translates into half a day as there is no detailed information in the methods as to how the EQ5D assessments, done on a weekly basis, were then translated into QALYs. The authors should provide an explicit example in the methods of how QALYs were calculated using AUC, and specify how the time horizon of 28 days was taken into account, and how 0.03 QALYs then translates into half a day. If the measurement difference is indeed 0.03 QALYs, then $0.03 * 365 \text{ days} = 10.95 \text{ days}$, not half a day. If there are space limitations, perhaps this could be included in the supplemental section.

The QALY was obtained for the 28-day time period (0,002). If we calculate $0,002 * 28$ and divide it by 365 we will obtain the QALY gain for the whole year.

Is there a reason no probabilistic sensitivity analysis was included? While the scenario analyses address some certainty associated with the base model, there is no test of uncertainty within each parameter used to generate cost and QALY estimates. Although the data are from a clinical trial, there needs to be uncertainty assessment of each parameter so that decision-makers considering this information in clinical practice can understand how different rates of utilization may affect the overall costs and consequences. The authors themselves state in the Discussion that “...the limited sample size combined with patient-level variation prevented a robust conclusion”; this is where a probabilistic sensitivity analysis would be helpful to characterize the amount of patient variation.

We agree this is true. However, no PSA was done. This is now stated as a limitation in the discussion section

It would be helpful to include a summary of the clinical trial results in the Methods section, first paragraph. Were statistical analyses done to control for smoking status and/or asthma / allergic rhinitis?

We considered that the methods section should only be related to the methodology used. The reference of the clinical results is provided in the Introduction section.

Were prescription costs for the placebo group included? Page 8, last sentence of top paragraph, says no cost was included for placebo medication. Are the authors referring to just the placebo tablets? If so, then please relabel placebo medication as placebo tablets, as "medication" implies therapeutic effect, which is not the case here. If placebo patients got prescription meds, those costs should be included, unless the number and type of prescription meds was identical between the treatment and control groups.

Placebo medication is meant to be placebo tablets. We consider that medication is clearly related to tablets only. However, we now clarified it the first time we use "trials medication" in the text, in order to emphasize it.

What "prescription medications" did patients get in addition to prednisolone / placebo? Did those in the placebo group get more rx medication because they went back for a repeat visit after their symptoms resolved? Nowhere do the authors discuss the other prescribed medications, or why only prednisolone was costed so precisely; was there no discount price and dispensing fee included in the cost of other rx medications?

The list of prescribed medications was very large and included many different drugs so no details were provided in the paper. Regarding the reasons for using the medications, no specific reasons were recorded during the study. Given the fact that the prednisolone dose used in the trial is not available in real life, a detailed calculation was made in order to get the accurate price of the intervention in the real setting

Table 2: why is there only a mean of 0.27 primary care consultations in the treatment group? Isn't a primary care consult required in order to get a prescription for prednisolone? Shouldn't the mean be closer to 1.0?

The prednisolone tablets were authorised right after the patients eligibility was approved. Therefore, this was not included as a GP visit in the analysis itself

The authors should include a table of the actual utility values, or at least a summary by week, comparing the two groups (a figure may be appropriate as well). This would help readers understand what (and when) is driving the differences in QALYs between each group. Table B provides no insight into the factors driving the differences between the groups.

We agree this is true. This is now stated as a limitation in the discussion section

It would be useful to include p-values in supplemental table B to compare whether there were significant differences between the two groups as far as % of patients reporting no problems.

We considered including the p-values unnecessary for this table, as the intention was to show the percentages for each group, not the difference between them.

Can the authors provide additional literature quantifying how the magnitude of their QALY estimates compare to other studies that look at similar populations? For example, although these patients were not necessarily on corticosteroids, the papers by Bergus et al, 2008, and Johnson FR et al, 2000, include QOL utility estimates that reflect respiratory illness and improvement in symptoms given treatment. Comparing the authors' results to these papers would provide better context for whether the results of the current study make sense.

The fact that these patients were not on corticosteroids makes the comparison not so relevant, in our opinion. Besides, references regarding quality of life and corticosteroids are already included in the paper.

Reviewer 2: Werkhoven (Netherlands)

I do not understand the QALYs value and the translation to a difference of "half a day in best imaginable health". A period of 28 days in perfect health makes a QALY of 0.077 ($1/365 \times 28$) so this is the maximum value of each participant. 0.03 QALY would translate to 11 days in perfect health (0.03×365). Otherwise stated, over the 28 day period, their quality of life is, on average, almost 40% better (absolute difference). This would be a major health improvement but I cannot imagine it is true! So please check the QALY calculation or enlighten this reviewer, maybe my own calculations are wrong.

The QALY was obtained for the 28-day time period (0,002). If we calculate $0,002 \times 28$ and divide it by 365 we will obtain the QALY gain for the whole year.

For scenario 5 (NHS perspective) and for the basecase scenario (patient perspective) it would add value if a willingness to pay curve was added to reveal what is a likely value of the cost per QALY gained. Many patients are able and willing to pay some money to recover faster, and in general society is willing to invest in QALY gain.

The inclusion of a cost per QALY indicator would imply including an additional analysis (cost-utility analysis) to be performed. In addition to that, it was not planned to perform such analysis in the protocol, that is the reason why we excluded it.

The authors performed a cost-consequence analysis. The difference with a conventional cost-effectiveness analysis is - as I understand it - that not only (quality of) life years gained but also other (clinical) consequences are reported. In this case, the duration of cough and the symptom severity score at day 2-4 post randomisation are reported. However, these outcomes were already reported in Hay et al. JAMA 2017, so although it might add value to report them again, it should be clearly stated in the abstract, methods and results section (including the tables) that these have been reported previously. If any difference, it should be clarified where they arise from.

The mention to the clinical effectiveness in the methods section is already linked to the reference in Hay et al., 2016.

In objective 'non-asthmatic adults' but in participants 'no asthma or other chronic lung disease'. Please use 'no asthma or other chronic lung disease' consistently.

Done

outcome measure: please include the time window of 28 days for the outcomes.

Done

"the benefit in terms of quality of life is small" -> did the authors have higher expectations in this population? Even if one would feel better two days earlier, that would probably still translate to "half a day in perfect health" (assuming severe cough would roughly reduce one's quality of life by 25%). What benefit would be large enough to justify one course of corticosteroids?

even when the benefit is reasonable, it is a small amount of QALYs gained and it should be stated like that, especially in a parameter such as the QALY, which can be comparable through different disease areas

In outcome measures: costs should also be included.

We agree, as this is an economic paper as well.

In strengths and limitations: 'The economic evaluation included the perspectives of patients and time off work as well as the NHS.' But from the abstract it seems the cost evaluation is only from a healthcare perspective.

Perspectives are described in the design section of the abstract

Page 8 line 28: please make clear that the EQ-5D-5L was also measured at baseline, maybe repeat the 28 day follow-up period.

Done

"Stata 13 and above" do you mean that you used higher versions as well? Please specify which versions.

The use of Software is usually specified using this sentence in papers. It is assumed that all versions from 13 onwards were used.

Please clarify whether "Prescription charge" is also included in the costs of non-study medication. (I think it should be.)

Yes, it is included

Page 10 line 10: Time off work was recorded by 321 (81%) participants. The wording might be clarified to avoid readers thinking the other participants were unemployed, or had zero work absenteeism, or did not complete this question. I assume the latter is meant, if so it might clarify if it is explicitly stated that 19% did not complete the question on work absenteeism.

Ok, we modified it in the text now

Page 10 line 11: Maybe to add what proportion had zero follow-up EQ-5D-5L questionnaires completed as this would severely complicate the area under the curve estimation.

I am not sure I understand this question. If this refers to missing data handling it was done in the sensitivity analysis scenarios.

page 12 line 22: "heath" should be "health".

Done

page 12 line 35-42: similar to previous comment, these data have already been reported. As they are not an integral part of the cost-consequence analysis (or else I have missed it) they should not be repeated. Of course they can be discussed in the discussion.

Despite this is data already published, it is part of the group of parameters related to health outcomes in OSAC. Given that this is a cost consequence analysis, all health outcomes must be addressed.

Table 4: Please also add summary statistics per intervention group. Please clarify that a HR > 1 means a shorter duration of cough. Please include the day of the symptom severity score.

Despite HR>1 indicates shorter duration of cough, this result is not statistically significant and therefore it should not be said that the duration of cough is shorter.

Table 5: Number of patients in scenario 5 has dropped off.

True, already stated now in the text

It would be valuable to have a detailed description of the other prescriptions provided. E.g. if prescribing prednisolone would prevent prescription of antibiotics it might be more worth it, despite the small difference in clinical outcome.

The list of prescribed medications was very large and included many different drugs so no details were provided in the paper

It would add insight if the QALY over the 28 day period would be plotted for the two groups.

Despite this is a fair point, the figure was not added given the small difference in QALYs obtained "Prednisolone was found to be clinically ineffective for treating ALRTI in non-asthmatic adults in primary care" Please make clear that this statement is about cough duration and symptom severity after 2-4 days (and maybe refer to the paper by Hay et al.)

Done

Reviewer 3:

Interestingly, this cost-consequence and patient-wellbeing analysis (also covers patients' wellbeing through the EQ5D questionnaire), has "positive" results through half a day earlier reached wellbeing in the prednisolone group, whereas the endpoints of the main manuscript were not significantly different (duration to resolving symptoms such as cough). I had to go back to the main manuscript to find this out. I therefore encourage the authors to highlight the difference between the main results of the RCT and this "alongside" analysis, i.e. predefined secondary analysis, in the discussion. Furthermore, I invite the authors to be more specific at the beginning of the discussion (page 13 line 21) : they state that prednisolone has been found to be clinically ineffective in ALRTI. They should state that they refer to the main manuscript of the RCT (please also include the reference there).

The cost effectiveness results regarding cough and mean symptom score are shown in the results section of this manuscript (before sensitivity analysis). Also, there is a reference to the results in the main manuscript (references 13 and 14).

I agree with the authors that the long-term risks of a "positive" glucocorticoid trial in a common disease should be highlighted, as it may result in a widespread frequent use of glucocorticoids. This main caveat has been well-covered by the authors. However, I encourage the authors to positively conclude the results of this analysis in the abstract, namely that the evaluation of economic aspects and patient-wellbeing resulted in a slight benefit "pro" glucocorticoids, before adding the caveat of potential harm.

This is a fair point. However, given the slight benefits obtained in terms of QALYs (0,03 QALYs) and the cost savings obtained by the prescription charges mainly, we chose to be conservative and considered more realistic to state that oral prednisolone cannot be considered a "beneficial" intervention for these patients.

I would like to encourage the editors to have a statistician review the paper.

No action needed from the authors

VERSION 2 – REVIEW

REVIEWER	Cynthia Gong University of Southern California Children's Hospital Los Angeles United States
REVIEW RETURNED	19-Aug-2019

GENERAL COMMENTS	Thank you for the opportunity to review a revision of this manuscript. Overall, there are still several critiques that have not yet been fully addressed. 1) I am still not convinced that the authors have adequately justified the choice of methodology (cost consequence analysis) when the standard in economic evaluation is to compare outcomes as cost per QALY or cost per alternate health outcome. Based on the results in Table 4, it would appear that prednisolone is in fact cost-effective due to decreased costs and increased QALY gains relative to placebo. I would change the title to "Economic Evaluation of Oral Corticosteroids..." because "Cost-Effectiveness" is misleading. 2) If it is too cumbersome to list out all other prescribed medications given in the trial, then perhaps drug classes could be listed. I notice that another reviewer also had the same concern, and it appears that this has not yet been addressed in this revision. 3) I would recommend moving the results of the clinical trial from "Results" to "Methods", because all reviewers expressed the same concern - that results of the trial were not made explicitly clear. This could be easily remedied and the manuscript would read more clearly if the trial results were presented upfront in the Methods section after the brief description of the clinical trial. The results section of this paper should present results of the economic evaluation, not of the clinical trial. 4) The authors have still not clarified how 0.03 QALYs translate into half a day. Both myself and another reviewer calculated $0.03 \text{ QALYS} * 365 \text{ days} = 11 \text{ days}$ in perfect health. Nowhere does it state that the QALY gain was 0.002 over 28 days. Based on the authors' response it is still unclear how this translates into half a day. If the QALY gain over 28 days was 0.002, then the calculation would be $(0.002 / 28) * 365 = 0.026$, and this translates into $0.026 * 365 = 9.49 \text{ days}$. The authors' response is $(0.002 * 28) / 365 = 0.00154$, which then translates into half a day ($0.00154 * 365$), but it is not clear why $0.002 * 28$ is the appropriate calculation in this case, unless the authors meant to say that the QALY gain was 0.002 per day in the 28 day period? In either case, the calculation should be made clear. 3) Can the authors add the PSA? I appreciate that the authors have added a sentence in the discussion about this, but this should be followed at the very least by a justification of why the PSA was deemed unnecessary in this case (similarly with a WTP curve). I would at the very least restate the discussion sentence to say: "Although we did not perform PSA, several scenario analyses were performed to account for potential uncertainty in the estimates". 4) In the summary of clinical trial results, please add that the rates of asthma medication receipt were the same as shown in Table A appendix. However, the authors have still not yet addressed whether COPD rates were the same in each group. This is an
---

	important confounder as COPD patients frequently require steroids for treatment! 5) Meaning of the study section - I think there needs to be much more discussion of how prednisolone use can offset abx use. True, it is unknown the possibility of side effects from repeated short term use, but abx resistance is such a huge problem that it seems that side effects from short course steroids (if any that are actually clinically severe) are minor in comparison. Even if cough duration and symptom severity may be a bit worse with prednisolone, isn't that already captured in the EQ5D assessments anyway, which show that despite this, there is an improvement in QOL?
--	--

VERSION 2 – AUTHOR RESPONSE

Reviewer:

Comment 1. I am still not convinced that the authors have adequately justified the choice of methodology (cost consequence analysis) when the standard in economic evaluation is to compare outcomes as cost per QALY or cost per alternate health outcome. Based on the results in Table 4, it would appear that prednisolone is in fact cost-effective due to decreased costs and increased QALY gains relative to placebo. I would change the title to "Economic Evaluation of Oral Corticosteroids..." because "Cost-Effectiveness" is misleading.

OSAC team response: we agree with the reviewer that the title should contain 'economic evaluation' rather than 'cost-effectiveness'; we have changed it now. Regarding the justification of the choice of methodology, we referenced Coast et al. 'Is economic evaluation in touch with society's health values?' BMJ. 2004 Nov 20; 329(7476): 1233–1236', which we consider the best source to date to justify and explain the reasons for choosing cost-consequence analyses. According to Coast et al. (2004), the 'Use of QALYs as a single outcome measure for economic evaluation means that important health consequences are excluded'. In OSAC there was a small QALY gain but no difference in duration of cough or severity of symptoms. When considering also the possible side-effects of corticosteroids, the short term effects may not compensate the long-term harms and a cost-consequence choice would show these implications in a better way than a cost-utility analysis.

Comment 2. If it is too cumbersome to list out all other prescribed medications given in the trial, then perhaps drug classes could be listed. I notice that another reviewer also had the same concern, and it appears that this has not yet been addressed in this revision.

OSAC team response: Despite limitations applied to accessing to the original data, the information we present in the paper suggests it was very small amounts, with limited impact from the economic perspective.

Comment 3: I would recommend moving the results of the clinical trial from "Results" to "Methods", because all reviewers expressed the same concern - that results of the trial were not made explicitly clear. This could be easily remedied and the manuscript would read more clearly if the trial results were presented upfront in the Methods section after the brief description of the clinical trial. The results section of this paper should present results of the economic evaluation, not of the clinical trial.

OSAC team response: we agree with the reviewer that this change would result in a clearer manuscript; we have moved the clinical results to the methods section in this revised version.

Comment 4: The authors have still not clarified how 0.03 QALYs translate into half a day. Both myself and another reviewer calculated $0.03 \text{ QALYS} \times 365 \text{ days} = 11 \text{ days}$ in perfect health. Nowhere does it

state that the QALY gain was 0.002 over 28 days. Based on the authors' response it is still unclear how this translates into half a day. If the QALY gain over 28 days was 0.002, then the calculation would be $(0.002 / 28) * 365 = 0.026$, and this translates into $0.026 * 365 = 9.49$ days. The authors' response is $(0.002 * 28) / 365 = 0.00154$, which then translates into half a day ($0.00154 * 365$), but it is not clear why $0.002 * 28$ is the appropriate calculation in this case, unless the authors meant to say that the QALY gain was 0.002 per day in the 28 day period? In either case, the calculation should be made clear.

OSAC team response: If the QALY gain was exactly 0.03 this would be 10.95 days of perfect health over a year ($0.03*365$). And over 28 days it would be 0.84 days (i.e. just over three quarters of a day). However, 0.03 is a rounded figure and a small difference can translate into a greater difference once it is changed to days. We tried to be conservative in saying the gain was "about half a day of perfect health" and we considered this sentence would make results more meaningful to the reader. We tried make this point slightly more explicit in this revised version of the paper. However, if the reviewer still thinks this is not clear, we will be very happy to delete the sentence and just keep the numbers.

Comment 5: Can the authors add the PSA? I appreciate that the authors have added a sentence in the discussion about this, but this should be followed at the very least by a justification of why the PSA was deemed unnecessary in this case (similarly with a WTP curve). I would at the very least restate the discussion sentence to say: "Although we did not perform PSA, several scenario analyses were performed to account for potential uncertainty in the estimates".

OSAC Team response: actually, PSA and WTP are most adequate when doing a cost-utility analysis, as they mainly refer to the cost-per-QALY results. Therefore we consider these type of analyses would not fit in our study and that is why they were not included or mentioned finally in the paper.

Comment 6: In the summary of clinical trial results, please add that the rates of asthma medication receipt were the same as shown in Table A appendix. However, the authors have still not yet addressed whether COPD rates were the same in each group. This is an important confounder as COPD patients frequently require steroids for treatment!

OSAC Team response: There were no patients with COPD at all included in the trial (please see supplement file attached). This is both at baseline (COPD was an exclusion criterion) but we also checked if any patients had been diagnosed in the three months following enrolment (in case response to the trial medication had prompted further investigation resulting in a new diagnosis).

Comment 7: Meaning of the study section - I think there needs to be much more discussion of how prednisolone use can offset abx use. True, it is unknown the possibility of side effects from repeated short term use, but abx resistance is such a huge problem that it seems that side effects from short course steroids (if any that are actually clinically severe) are minor in comparison. Even if cough duration and symptom severity may be a bit worse with prednisolone, isn't that already captured in the EQ5D assessments anyway, which show that despite this, there is an improvement in QOL?

OSAC Team response: we agree with the reviewer this is an important point; any interventions that could reduce antibiotic use would be relevant; however we consider this point is already addressed in the "unanswered questions and future research"; besides, we looked and did not find any difference in antibiotic consumption between the two groups (please see Table 3 of the Clinical paper attached). We have concluded that prednisolone does not benefit acute LRTI disease specific outcomes, though it does (as has previously been observed) have positive non-specific effects on well-being. We do not consider these non-specific benefits warrant the short and long term side effects, particularly if treatment became so popular it resulted in individuals having repeated courses.

On the other hand, we believe the EQ5D could maybe not capture the effects of LRTI and those of prednisone/antibiotics on the patient adequately, as some domains of the questionnaire would not be relevant for this disease and its treatments.